# Transformation of the Linji School's Perspective on Seated Meditation from Tang to Song Dynasties—From Negation to Returning of Seated Meditation

**Jing-Jia Huang**

Department of Chinese, National Taiwan Normal University, Taipei 106, Taiwan; elenna@ntnu.edu.tw

**Abstract:** This study first examines the developmental perspective on seated meditation from Bodhidharma 達摩 to the Linji 臨濟 school of the Tang Dynasty. During this dynasty, the Linji school followed the criticism of physical seated meditation by the Southern school of Huineng 惠能 and mainly used wisdom dialogues to enlighten Buddhist disciples. Furthermore, there are very few lamp records documenting the seated meditation of Chan masters in the Linji school, which has created the impression that this school in the Tang Dynasty disregarded physical seated meditation and even negated it. Second, this study examines the attitude towards seated meditation of Linji Chan masters during the Song Dynasty from two aspects of seated meditation, namely, practice and theory. It is found that their attitude towards it differed from that of the Linji school in the Tang Dynasty because they were usually experienced in seated meditation. Moreover, there were Chan masters in both the Yangqi 楊岐 and Huanglong 黃龍 schools that advocated seated meditation as the foundation of Buddhist practice. For example, both Foyan Qingyuan's 佛眼清遠 "Seated Meditation Inscription" 坐禪銘 and Foxin Bencai's 佛心本才 "Seated Meditation Etiquette" 坐禪儀 pay attention to seated meditation and agree that enlightenment can only be achieved through gradual cultivation, and after enlightenment, a period of training is still required to achieve ultimate perfection. The above findings suggest that there was a development tendency of returning to seated meditation in the Linji school during the Song Dynasty.

**Keywords:** seated meditation; Linji school; Chan practice; Chan method; Chan school; Tang and Song Dynasties

## 1. Introduction

The meaning of the phrase "perspective of seated meditation" includes the proposition, interpretation, and attitude toward seated meditation as well as the overall perception of the method and process of practicing it. Based on the actual practice history of Chan schools, Chan masters in China add their personal practical experience to guide Chan practice, rather than follow the guidelines for seated meditation prescribed in Buddhist classics, which is the main reason the Chan practice method is constantly changing in China over time. Even with the same Chan practice method, the nuances of the seated meditation practiced by different Chan masters can differ. Therefore, different perspectives and practice methods of seated meditation marked by varying individual or school characteristics have been developed. Regarding the classics of Chan Buddhism, Chan masters' instructions regarding Chan practice and Chan master–disciple wisdom dialogues presented in the series of translations by the outstanding contemporary Buddhist translator and scholar Thomas Cleary are worthy of reference.[1]

Huineng's 惠能 (638–713) southern school has always conveyed the impression that this school opposes seated meditation. This impression originated from Huineng's redefinition of "seated meditation". Nanyue Huairang 南嶽懷讓 used the example of "grinding a brick into a mirror" to enlighten Mazu Daoyi 馬祖道一, Baizhang Huaihai 百丈懷海

emphasized enlightenment in daily life, and Linji Yixuan 臨濟義玄 mainly used wisdom dialogues to enlighten Buddhist practitioners and thus developed the Linji school, which does not include physical seated meditation and even negates it. Did the school's position and attitude regarding seated meditation change in the Song Dynasty? Song society experienced a period of transformation when the cultural trend gradually shifted from the aristocracy of the Middle Ages to the modern urban civilization, in which economic development, the rise of the middle class, and the advancement of printing technology quickly promoted cultural changes. Owing to the influence of adapting to the trend of the new era, the way Chan schools developed differed from the way they did in the Tang Dynasty. Albert Welter, a contemporary Chan researcher, has pointed out that our impression of the Linji school in the Tang Dynasty reflects the image of a Chan master constructed through the "Linjilu", which was edited in the Song Dynasty (Welter 2008). We can only consider the historical background of the text as much as possible to understand the development of and changes in the Linji school's concept of meditation in the Tang and Song Dynasties, as presented in the text.

Compared with the Caodong school 曹洞宗, which followed the practice traditions of the mountains and forests, the Linji school was more capable of adapting to changes in the environment of the times, and hence, it became the most active and influential school in the Song Dynasty. Two sects of the Linji school, namely, Huanglong and Yangqi, developed in the Northern Song Dynasty. Based on the historical context presented above, this study first examines the Linji school's perspective of seated meditation before the Tang Dynasty. Then, it examines the differences and changes between the perspectives of seated meditation in the Song and Tang Dynasties from the two aspects of seated meditation: the practice and theory of the Linji school Chan masters in the Song Dynasty.

## 2. Development of the Linji School's Attitude toward Seated Meditation before the Tang Dynasty

Meditation is the common foundation of all Buddhist traditions. The teachings on meditation practice methods, encompassing "Samatha" 止 and "Vipassana" 觀, have been propagated from Āgama scriptures. Owing to the spread of Buddhism to the East, the Mahayana and Theravada meditation texts were disseminated to China during the Eastern Han Dynasty. Examples are the "seated meditation breath-counting method" from *Mahayana Ānāpānasmṛti Sūtra* 大安般守意經, translated by An Shigao, belonging to Theravada meditation methods; and the *Seated Meditation Samadhi Sutra* 坐禪三昧經, translated by Kumārajīva, which contains both Mahayana and Theravada meditation methods and integrates Prajna thoughts. On the other hand, in the early development of China Chan Buddhism, the term "Chan Master" was different from "master explaining the meaning"; it was a title with a clear practical meaning and referred to another type of person engaged in the practice and teaching of meditation. Therefore, Chan schools were established based on "Chan", and the schools were characterized by their clear focus on the practice of Dhyana.

The image of the "Wall Viewing" of the Chan school's first monk, Bodhidharma, and the "Two Entrances and Four Practices" 二入四行, which represents the Mahayana Wall Viewing Meditation technique that he bequeathed to his disciples, differed significantly from the Mahayana and Hinayana that spread to China in the early days, which taught people how to practice seated meditation breath-counting. Bodhidharma focused on "peace of mind", and his mind-concentration technique of wall viewing was not only physical seated meditation, but also the accumulation of four practices. This change in the focus of meditation was the source of the southern school's approach of emphasizing the practice mode combined with daily life in the future.

The Fourth Patriarch Daoxin 四祖道信 attached great importance to the accumulation of meditation skills. *The Jingde Record of the Transmission of the Lamp* 景德傳燈錄 mentions that he "practiced Sesshin and never lay down to sleep for 60 years". He advocated the following in *The Suitable Method of Attaining Contemplation and Enlightenment of Mind*

入道安心要方便法門: "If one only practices Samatha for a long time, his mind will easily feel dull; if one only practices Vipassana for a long time, his mind will easily become distracted". 常住於止，心則沉沒；久住於觀，心則散亂。 Therefore, those who place too much emphasis on either Samatha or Vipassana will experience their respective disadvantages, while those who are untroubled by peacefulness and chaos can become Buddhist practitioners of "Chan and attentiveness". Moreover, Daoxin emphasized that, as there is no other Buddha outside the mind, it is most important to cultivate the "mind". In addition, he mentioned the method and key points for "beginners of seated meditation". "Dongshan Dharma 東山法門", as established by the Fifth Patriarch Hongren 五祖弘忍, emphasize that, since Bodhidharma, seated meditation is the foundation of practicing Buddhism, and it attaches importance to the practice mode that extends the concept of meditation to life as a whole. The importance of seated meditation has gradually been downplayed, while the status of "mind" has been accentuated. Moreover, the cultivation of the mind in daily life has become the focus of Chan practitioners' practice, while the importance of seated meditation has gradually been downplayed.

Shenxiu's 神秀 (606–706) northern school particularly emphasizes that its seated meditation tradition is inherited from Dongshan Dharma. Buddhist practitioners can eliminate delusion and troubles through seated meditation, and the Buddha nature will naturally appear following the long-term practice of seated meditation. Contrary the northern school, which emphasizes seated meditation, in the Biography of Master Caoxi, 曹溪大師別傳, Huineng received this instruction from Hongren: "There is only a need to pay attention to seeing the true nature, and there is no need to emphasize that meditation can lead to liberation" (Unknown author n.d. CBETA, X86, p. 50c). In other words, Huineng prioritized "seeing the true nature". The previous emphasis on the sequence of liberation from meditation was ignored, and instead, the importance of being open to self-nature and wisdom was emphasized. The practice method was changed owing to this transformation of the core concept. The Huineng southern school redefined "seated meditation" in this form: "Outside all good and evil realms where the mind is not involved, it is called sitting; seeing the self-nature without any change, it is called Chan". The southern school—which transcended the obsession with the seated meditation of the body, eliminated the formal boundary between sitting up and sitting down, and emphasized awareness of the mind—criticized the northern school's seated meditation, which attaches importance to the outer appearance, stating, "Some masters teach seated meditation by asking Buddhists practitioners not to move their body, so that their mind can be purified. This is the inversion of obsession" (Fahai n.d., CBETA, T48, p. 339a).

The change of Huineng's attitude toward seated meditation had a profound impact on Chan Buddhism after Nanyue and the Linji school. Nanyue Huairang used the example of "grinding a brick into a mirror"[2] to teach Mazu that "If you practice seated meditation, the focus of meditation is not on sitting. If you imitate a sitting Buddha statue, Buddha has no fixed form". He further added, "If you insist on sitting, you fail to understand the true meaning of Chan". Thus, Nanyue Huairang bequeathed Huineng's negation of physical seated meditation for Chan practice. Mazu Daoyi inherited the idea of "negating seated meditation" and pointed out that all living beings "have an inherently pure nature, and they do not need to practice seated meditation to obtain it. This is the pure meditation of the Tathagata". 本有今有不假修道坐禪，不修不坐即是如來清淨禪。 (Daoyuan 1004, CBETA, T51, p. 440b). Mazu Daoyi further combined practice with life, arguing that "self-mind is Buddha" and that the "ordinary mind is the way to practice Buddhism" and advocating a way to practice with purity of the mind in the moment, i.e., eating and sleeping without neglecting self-nature.

Baizhang Huaihai attached great importance to enlightenment in daily life, while Linji Yixuan used wisdom dialogues to enlighten Buddhism practitioners, which contributed to the Linji school's attitude of disregarding or even negating physical seated meditation. Moreover, Linji Yixuan inherited the tradition of criticizing seated meditation after the southern school and said, "Some ordinary Chan monks practice seated meditation when



they are full and do not let go of their thoughts. They hate noise and pursue tranquility, which is heresy" (Huiran n.d., CBETA, T47, p. 499b). This statement seems to belittle the northern school, which emphasizes the practice of seated meditation to seek purification as well as that of Sesshin to seek stabilization.

Japanese scholar Prof. Hiroo Shiina suggested that the northern school's perspective of seated meditation contains awareness of the practice that has been passed down since Bodhidharma, while the southern school has consciously established its particular characteristics by denigrating the tradition of seated meditation (Shiina 1971, pp. 134–46). In particular, in Heze Shenhui's 神會 *Determining the Right and Wrong of the Southern School based on Bodhidharma* 菩提達摩南宗定是非論, the exposition of the early Chan patriarchs deliberately omitted content about seated meditation in an apparently strategic piece of writing against the northern school. From the viewpoint of school relations, American scholar John R. McRae pointed out that the discourses on Chan history in both the northern and southern schools conceal strong school narratives and rhetorical power, which are irrelevant to literature. To a certain extent, the southern school's opposition to seated meditation may have been a development strategy to differentiate itself from the northern school (McRae 2004, pp. 1–21).

However, were the southern and northern schools' attitudes toward seated meditation completely opposite? In fact, Dongshan Dharma attached great importance to seated meditation as the basic training for Buddhist beginners, which does not fundamentally conflict with the extension of meditation to daily life. Therefore, the emphasis placed on the practice of Buddhism in daily life should not be interpreted as opposition to seated meditation. Master Yin Shun's *Chan Buddhism in China* also pointed out that there are many records of Chan masters practicing seated meditation in the Hongzhou school 洪州 and Shitou school 石頭 throughout the history of Chan (Yin Shun 1994, pp. 350–51). I also found examples where the Linji school Chan master's perspective of seated meditation contradicts their practice; for example, Mazu's disciple, Ehu Dayi鵝湖大義, who wrote "Seated Meditation Inscription", clearly inherited Mazu's stance against physical seated meditation. He said, "I deeply lament that seated meditation resembles a dead person, and it is only similar to this for a thousand years. If you regard this as a Chan school, you will lose the tradition of Shakyamuni Buddha of holding flowers and smiling" (Rujin n.d., CBETA, T48, p. 1048b). However, the "Inscription of Chan Master Dayi" says that he once "set up a tin stick and [practiced] seated meditation for three days". Apparently, in terms of practice, he spent a long time practicing seated meditation. Therefore, some scholars have reflected on content regarding seated meditation in the regulations of the Chan temple and have suggested that long-term seated meditation was still practiced in the Chan temple (Foulk 2004, pp. 275–312).

### 3. The Seated Meditation Practice of Chan Masters in the Linji School in the Song Dynasty

Chan school records mainly documented Chan questions and answers between the Chan master and his disciples; they rarely documented the Chan practice of the Chan master himself. However, records of the seated meditation practices of Linji school Chan masters during the Song Dynasty can still be found in biographies of monks or quotations from Chan masters. For example, *Biographies of Samgha Treasure of the Zen School* 禪林僧寶傳 Volume 3, the "Biography of Shanzhao" records the following about Fenyang Shanzhao 汾陽善昭 (947–1024) in the prince's temple: "He sat quietly on the Chan mat and never stepped outside for 30 years". Apparently, he practiced seated meditation for a long time. It is noteworthy that, in the Dharma lineage, despite Shanzhao being in the Dharma lineage of Linji school's Shoushan Xingnian 首山省念 (926–993), he "preferenced for the Caodong Chan method" and wrote the "Caodong Five Buddhist hymn", which was used by the Caodong school for guidance. Ciming Chuyuan 慈明楚圓 (986–1039), Shanzhao's disciple, mainly played an intermediary role in the inheritance of Chan, and he laid the foundation for the Linji school to spread to the south of the Yangtze River. While Chuyuan inherited

the Linji school's attitude of opposition to physical seated meditation in terms of ideology, his actual Chan practice still attached importance to seated meditation. Thus, there was a contradiction between concept and practice.

Chuyuan's disciples Yangqi Fanghui 楊岐方會 (992–1049) and Huanglong Huinan 黃龍慧南 (1002–1069) established the two most popular sects in the Linji school in the Song Dynasty, Yangqi and Huanglong, respectively, which influenced the development trend of Chan in the entire Song Dynasty. *Jia Tai Pu Lamp Records* 嘉泰普燈錄, Volume 3, records that Huanglong Huinan once said, "Those who become monks must express the ambition of a man of great courage, cut off the two ends, and return home to sit firmly, so as to open the door to benefit human beings". In other words, he advocated that monks must first achieve seated meditation before helping sentient beings. Huinan himself practiced seated meditation as well. In *Biographies of Samgha Treasure of the Chan School* Volume 22, "Chan Master Huanglong Huinan", he is recorded as the following: "He always sits in a lotus position and looks directly when walking". *Forest Groves Record* 林間錄 Volume 1 records that, when he was living in Guizong Temple in Lushan, there was a fire, due to which the crowd made noise in the valley, while Huinan sat peacefully as usual. "He was as immovable as a mountain". *Jia Taipu Lamp Records* Volume 6, "Chan Master Sixin Wuxin 死心悟新", records Wuxin's enlightenment: "One day, after practicing seated meditation silently, when he was sitting down, he was suddenly shocked when he heard the governor appease the practitioner, and he became enlightened. When he met the master, he even forgot to put on his shoes. He said to himself, 'All [the] people in the world are only practicing Chan, but I have realized Chan.'" Apparently, he attained enlightenment by comprehending the meaning of Chan through "the silent practice of seated meditation". Therefore, "seated meditation before investigating a meditation topic" aids in enlightenment.[3] According to these scattered historical materials, seated meditation remained an important daily practice activity for Chan masters of the Huanglong School. *Yun Wo Ji Tan* 雲臥紀談 records that, although Foxin Bencai 佛心本才 studied Chan diligently at the end of the Northern Song Dynasty, he still failed to achieve enlightenment. Afterwards, he achieved enlightenment by reading "Returning from the Mountain of Medicine" in the *Caodong Full Records* 曹洞廣錄. In other words, his enlightenment was inspired by Caodong Chan, and the core of the Caodong school's practice has always attached importance to seated meditation.

Yangqi Fanghui's Chan method is flexible, adaptable, and does not follow a fixed model, as he mainly inherited the practice tradition of the Linji school, which emphasized the practice of Chan in daily life, but disregarded the form of Chan practice. *Yangqi Fanghui Monk Quotations* records that he once went to the hall and said, "Stick to sitting until the universe disappears, the world will be dark; letting go of the obsession, the rain will stop, and the sun will shine again. Even so, there are still habits that have not been eliminated". 坐斷乾坤，天地暗黑；放過一著，雨順風晴。雖然如是，俗氣未除在。 (Renyong n.d., CBETA, T47, p. 640c). In other words, he believed that, if one is obsessed with seated meditation, he may not be able to see his self-nature even if he integrates himself into the universe. If one can let go of the obsession with seated meditation, enlightenment will follow according to the right timing and cause. This passage describes an adjustment of the process of the seated meditation practice. "Letting go of the obsession" did not necessarily mean that he opposed seated meditation. Rather, it meant that, after "Stick to sitting until the universe disappears", he naturally let go of his obsession with physical seated meditation and entered a higher level of the realm. At this moment, while his inner meditation was stable, there were still personal inertia and habits. Therefore, the two realms mentioned by Fanghui were not opposite, and his words should not be interpreted as negating seated meditation.

When the Yangqi school was passed down to Yuanwu Keqin 圓悟克勤 (1063–1135) in the middle and late Northern Song Dynasty, its popularity gradually surpassed that of the Huanglong school. He once said, "After the Buddha was born, he merely taught to cessation knowing". Afterwards, you will achieve the goal of "walking is walking, sitting

is sitting, wearing clothes is wearing clothes, and eating is eating, which is similar to a bright mirror where everything is reflected as it is; without the awareness of discrimination, self-nature can be seen everywhere in daily life" (Shaolong 1133, CBETA, T47, p. 773b). According to his philosophy, the Chan practice is to let go of all discriminations and perceptions, and our self-nature is like a mirror that has existed since time immemorial and can manifest itself in response to the situation without obstacles. Therefore, practice can be completely combined with daily life without distinction until the point "where there is no thought, you can see the true nature". Keqin's disciple Dahui Zonggao 大慧宗杲 (1089–1163) advocated "Kanhua Chan" 看話禪, and believed that, by focusing on the study of "Huatou" 話頭 and eliminating the distracting thoughts of consciousness, one can reach the state of no thoughts. This process of enlightenment can be achieved using seated meditation or while standing, sitting, or lying down; it has no limitation.

The close interaction between Caodong and Linji continued from the Five Dynasties to the Song Dynasty. Prof. Masahiro Hasegawa once investigated the close interactions between the Chan masters of the Buddha Eye Sect and the Huqiu 虎丘 Sect of the Yangqi and Caodong schools. Since the styles of all three schools were simple and plain and they all attached importance to seated meditation and a life of rigorous practice, there were close interactions and a profound relationship between them from the end of the Northern Song Dynasty to the Southern Song Dynasty (Hasegawa 1992, pp. 273–79). This would also be one of the reasons the Buddha Eye Sect and the Huqiu Sect of the Linji school returned to seated meditation at the end of the Northern Song Dynasty.

## 4. The Seated Meditation Theory of the Linji School in the Song Dynasty: Returning to Seated Meditation, as Based on Foyan Qingyuan's "Seated Meditation Inscription" and Foxin Bencai's "Seated Meditation Etiquette"

*Jia Taipu Lamp Records*, Volume 30, "Miscellaneous Collections", collected Foyan Qingyuan's "Seated Meditation Inscription" and Foxin Bencai's "Seated Meditation Etiquette". Both Foyan Qingyuan and Foxin Bencai practiced Chan under the guidance of Cimin 慈明 in the Linji school. The former belonged to the Yangqi School, while the latter belonged to the Huanglong School. As they lived during approximately the same time around the end of the Northern Song Dynasty, they are regarded as representatives to understand the changes in the Linji school's perspective regarding seated meditation in the Song Dynasty:

(1)    In the Yangqi School, Foyan Qingyuan's "Seated Meditation Inscription"—Affirming that there is no difference between movement and stillness in achieving enlightenment through seated meditation.

Foyan Qingyuan 佛眼清遠 (1067–1120) was a disciple of Wuzu Fayan 五祖法演 (Uncertain birth–1104). According to the descriptions in Li Mi-Sun's 李彌遜 "Chan Master Foyan Tower Inscription" 佛眼禪師塔銘, Qingyuan was a Buddhist practitioner who "was strict and quiet, spoke little, and laughed and moved with rules. Qingyuan first lived in seclusion in the Dazhong Temple, then became the abbot of the Longmen Temple. He was famous for his Buddhist instructions, and scholars were attracted to learn Buddhism from him. He had a strict attitude when teaching his disciples, did not like to give empty talks, and never easily gave recognition to a disciple who was proficient in Buddhism unless there was solid evidence" (Ze zang zhu n.d., CBETA, X68, p. 227c). In order to rectify the Chan practitioners' blind pursuit of the plain Chan style at the time, he paid special attention to the state of enlightenment and attached great importance to seated meditation. Although Huqiu Shaolong 虎丘紹隆 (1077–1136) inherited the Dharma from Yuanwu Keqin, he was closer to his master's fellow master, Foyan Qingyuan, who valued the Chan style of seated meditation. Shaolong's disciple Yingan Tanhua 應庵曇華 (1103–1163) wrote "Instruct Zhang Xiu Zao" in his collection to summarize the two key points of Qingyuan's Chan style: First, he praised Foyan Qingyuan for "enlightening the world with Dongshan Dharma", omitting Huineng's southern school and directly narrating Dongshan Dharma. However, from Daoxin's to Hongren's time, attention was still paid to the practice of seated

meditation. Second, he commented that the Chan practice of Qingyuan was the integration of actions and principles: "Although it is not similar to the Linji school Chan masters who scold and hit disciples to teach Buddhism, and neither does it use stick and sharp warning, its influence is the same as the Linji school Chan masters who scold and hit disciples to teach Buddhism" (Suochuan 1166, CBETA, X69, p. 535b).

The content of Qingyuan's "Seated Meditation Inscription" does not include detailed guidance on the method of seated meditation, but mainly discusses the spiritual state that seated meditation can achieve and exhorts the practice of seated meditation, which is also in line with the function of the "Inscription" for admonition and exhortation. Both Prof. Ishii Shudo and Prof. Kyushi Seito believed that Qingyuan's concepts initially explained that, "movement and stillness are always in samadhi" 動寂常禪 and "let the self-nature operate freely, without pretense, without thinking and distinguishing" 任運滔滔 are to advocate that meditation can be practiced when walking, standing, sitting, and lying, without being obsessed with the sitting posture. In other words, Qingyuan inherited the Huineng Chan master's "liberation from physical seated meditation", which is consistent with the Koan of "grinding a brick into a mirror". They believe that Qingyuan opposed physical seated meditation and inherited the concept to eliminate the obsession with it (Ishii 1997, pp. 51–53; Seito 2007, pp. 47–48).

However, based on the context, I have different interpretations:

(1) The beginning of the text explains the ultimate state of "seated meditation": "The essence of the mind is emptiness, and its nature does not differentiate between deviation and circle", 心元虛映，體絕偏圓. This is an explanation of the original purity and original state of the "mind", as well as the goal of meditation. If one can achieve this state, he will not leave the samadhi even when he is sitting or no longer sitting. Moreover, his "movement and stillness are always in samadhi", and he will no longer be obsessed with the distinction between sitting and no longer sitting. However, this state must be achieved after "the rising and extinguishing of thoughts [that] are all quiescent, and the state of samadhi is achieved" 起滅寂滅，現大迦葉. Then, one can achieve what the Linji Patriarch described saying, "Sitting, lying, and moving are all kept in samadhi" 坐臥經行，未嘗間歇. In other words, the key is to achieve deep samadhi; provided one can achieve deep samadhi, he does not have to be obsessed with the sitting posture. At this time, if one is still obsessed with the external sitting posture, this obsession will become a "dharma obstacle", which is a form of obsession with "using the Buddha to find the Buddha". On the contrary, if one has not achieved deep samadhi, the concept of "movement and stillness are always in samadhi" cannot be fulfilled. Therefore, Qingyuan affirmed that, when the mind returns to its original purity and original state, there is no difference between sitting and no longer sitting. However, this is under the premise of achieving the state of deep samadhi. The above suggest that his attitude toward meditation aligned with that of "Dongshan Dharma", which emphasizes the foundation of Chan meditation and differs from the attitude of the Linji school, which emphasizes that daily life is practice.

(2) A long paragraph discussing the psychological changes associated with seated meditation follows this text, and it is often ignored. Hence, if we wish to understand Qingyuan's position regarding seated meditation, we must read the full text. In the second paragraph, Qingyuan spoke about the psychological changes that occur in beginners of seated meditation. Since their minds have not been trained, it is impossible for them to immediately see their self-nature; therefore, they need to go through the method and training of "seated meditation". "From the very beginning sat upright, the mind was full of distractions" 初心鬧亂. This refers to a delusional mind that has not been trained and is without awareness. After a long time, "the six senses will no longer respond to the external environment", 虛閑六門. Even when the senses engage disparate thoughts, these

thoughts themselves remain impermanent. As long as there is the awareness that thoughts originate from your own mind, the reactions of the six senses will gradually stop, and the thoughts will gradually become clear, i.e., the "mind becomes unobstructed", 心心無礙, which transcends the binary system. Thus, this paragraph provides a detailed explanation of the psychological changes that beginners at the ordinary level will experience when they approach the original pure state of mind through seated meditation training. Additionally, it clarifies how to eventually achieve the ultimate liberation to make "life and death never exist" 生死永息. Apparently, Qingyuan affirmed that seated meditation is the foundation of the Chan practice. Prof. Masahiro Hasegawa's view differs from that of Ishii Shudo and Kyushi Seito, as he agrees with the clear position of Qingyuan's "Seated Meditation Inscription", which emphasizes seated meditation (Hasegawa 1993, p. 12).

(3) The final paragraph encourages the Chan practice and emphasizes that the difference between mortals and saints lies in enlightenment.

In terms of ordinary people's seated meditation, Qingyuan proposed that, when the state of deep samadhi is achieved through seated meditation, it aligns naturally with the state explained by the statement "movement and stillness are always in samadhi". If one achieves a state wherein there is no difference between sitting or no longer sitting, there is no need to be obsessed with seated meditation; therefore, seated meditation is the means, not the end. The end of the text reemphasizes that the core of the seated meditation practice lies in "enlightenment" to see the pure self-nature, and it exhorts practice to attain enlightenment. In terms of the coherence of the context, the overall inscription does not invalidate physical seated meditation. On the contrary, Qingyuan attaches importance to the key role of seated meditation in the process of realizing one's self-nature.

Qingyuan's other text "Revealing the Essentials of Mind to chan Practitioners" 示禪人心要 also emphasizes that "If one can sit still and calm his mind, he will naturally gain strength over time". In other words, one can naturally "obtain tranquility in the midst of chaos" and "all worries are Bodhi".[4] In other words, when one achieves Buddha's samadhi—nāga—[5] through seated meditation, he can achieve the state wherein movement and stillness are always in Samadhi. Afterward, he can let his thoughts arise and cease as he wishes. By taking Qingyuan's personality, the Buddhist learning style, and the sect relationship into account, this study found that he was inclined to the Caodong's plain Chan style, which affirmed seated meditation, although he was in the Yangqi school.

Moreover, the Qingyuan Quotations read: "If one wants to achieve the original purity of his mind, he needs to go through the process of seeking and visiting a master, studying day and night, and cultivating his mind until he achieves enlightenment. Afterward, he will realize that he never lost his pure mind, even when he had not aroused his mind. The Patriarch Maming 馬鳴 called this 'beginning enlightenment' 始覺, which is 'original enlightenment,' 本覺 as there is no difference between beginning enlightenment and original enlightenment, and they are collectively called the 'ultimate enlightenment' 究竟覺". (Ze zang zhu n.d. CBETA, X68, p. 217c).[6] Qingyuan quoted *Awakening of Faith in the Mahayana* 大乘起信論 to explain the process of achieving the state of the original pure mind from the concept; this means "beginning enlightenment" will eventually lead to "original enlightenment". Apparently, Qingyuan agreed that enlightenment can only be achieved through the process of gradual accumulation. It is not achieved all at once, but through stages of meditation.

(2) In the Huanglong school, Foxin Bencai's "Seated Meditation Etiquette"—Affirming seated meditation and gradual accumulation

Foxin Bencai 佛心本才 (dates of birth and death unknown) was a disciple of Lingyuan Weiqing 靈源惟清 (d.1117), and he died around the Shaoxing period. Hence, he lived in approximately the same era as Foyan Qingyuan. The content of Bencai's "Seated Meditation Etiquette" is as follows:

(1) The beginning of the text explains the level transformation of seated meditation and mind cultivation: When performing seated meditation, one should first "correct his thoughts" 端心正意 and adjust his motivation; the body posture should be "stacking feet and sitting cross-legged" 疊足加趺. Then, he should "watch and listen to his inner world", 收視反聽, that is, withdraw his gaze from the outside to the inside to maintain awareness. In addition, he should "stay away from drowsiness and restlessness", 沉掉永離 that is, any distracting thoughts should be ignored, and a state of mental tranquility should be maintained. The above step is similar to practicing "stopping". Second, after mental concentration and stability is achieved, use the calm mind to maintain awareness, in which state one is "aware of the present mind and reflecting on the present mind" 知坐是心，及返照是心. "This mind is empty, aware, still, and clear", 此心虛而知、寂而照 and does "not fall into the state of extinguishing and eternity" 不墮斷常. The above part is similar to practicing "seeing".[7] Therefore, Bencai's guidance on seated meditation includes the concept of first practicing stopping and then practicing seeing. He further pointed out that the focus of seated meditation is to adjust the "mind" from contemplating the present mind in reality, which is a thought biased toward the experiential level. Being aware of and reflecting on the present mind is similar to what is described in Qingyuan's "Seated Meditation Inscription": "Use one's own mind to reflect on it again" 還用自心，反觀一遍. This "mind" is peaceful and always able to achieve enlightenment, and when enlightenment is achieved, one can always maintain awareness, i.e., can see the original Buddha nature.

(2) The text subsequently points out that the problem faced by ordinary people who "focus on the practice of seated meditation" but "fail to achieve enlightenment" is that their mind clings to the outside, and they mistakenly believe that there are independent existences outside the mind. Thus, they are obsessed with the physical existence of the self and the things perceived by the self. As they cannot comprehend that the core of seated meditation lies in the enlightenment of the "mind", they lack the inward and spontaneous motivation to practice meditation. The text then proposes the correct attitude of seated meditation: If one can focus on one thoughts inwardly and become aware that his mind is originally pure, he will never experience the cycle of life and death, all fetters and unclearness will melt away in an instant, and he will be able to see the reality of all dharmas beyond birth and death. In other words, he can affirm that "his mind is a Buddha, and there is no Buddha outside of his mind". 自心外無別佛.

(3) Following the extinguishment of all the troubles of kalpas and unclearness mentioned in the previous paragraph, the description "continuing to practice while comprehending the inherent purity of the inner being, one will eventually attain the ultimate Buddhahood" 順悟增修，因修而證 refers to the further level of practice after realizing that the self-mind is a Buddha. After attaining samadhi, one can proceed on "the path of an-ābhoga"[8] 無功用道 to achieve the way of freedom without artificiality. Consequently, one can freely improve wisdom and merit. As the *Vimalakirti Sutra 維摩詰經* says, "Without leaving the samadhi, one can also embody the four majesties, which is the real meditation". 不起滅定而現諸威儀，是為宴坐。 In other words, a person who sees nature can adapt to the external environment and benefit all beings freely, and all walking, standing, sitting, and lying down are nothing but the display of the Buddha nature. This "sitting in meditation" refers to Mahayana meditation, which is neither fixed nor uncertain: even in walking, standing, sitting, and lying down, one is inseparable from samadhi. Therefore, there is no need to leave samadhi to be able to display the four majesties, and reaching this level of "seated meditation" conforms to Huineng's high-level definition of "seated meditation", which can completely surpass the constraints of external seated meditation.

Bencai's "Seated Meditation Etiquette" emphasizes the importance of seated meditation and is based on the belief that the awareness of self-mind as well as self-knowledge of the mind originating from the initial rectification of the mind is the Buddha, and one can achieve "specializing in one practice and attaining samadhi" 一行三昧. When one attains the Eighth Bhumi bodhisattvas, he can benefit all living beings freely. Apparently, this whole process of practicing meditation is not enlightenment attained all at once but has a clear sequence of practice and different levels of awareness. Prof. Ishii Shudo's "Textual Criticism on Seated Meditation Mottos" also points out that Bencai's "Seated Meditation Etiquette" contains the concept of "sudden enlightenment followed by gradual cultivation" 頓悟漸修, as proposed by Zongmi (780–841) 宗密 (Ishii 1997, pp. 55–56).

Zongmi's "Preface to the Collection of Various Writings on the Chan Source" 禪源諸詮集都序 advocates the practice sequence of "After sudden enlightenment, there is still a need for gradual cultivation". From Zongmi's perspective, "Enlightenment" has two levels: "If one practices because of awakening, this is enlightened understanding; and then if one becomes enlightened through practice, this is enlightenment. 若因悟而修，即是解悟；若因修而悟，即是證悟。 All the points above are only limited to this life; if we return to the accumulation of many lives in the past, it is all enlightenment through gradual cultivation, not sudden enlightenment. 然上皆只約今生而論，若遠推宿世，則唯漸無頓。 Sudden enlightenment in this life is achieved through the accumulation of gradual cultivation in many past lives". 今頓見者，已是多生漸熏而發現也。 (CBETA, T48, p. 408a02). In other words, the process from enlightened understanding to truth enlightenment must be connected by actual practice. The so-called "enlightenment" means "enlightenment through gradual cultivation". This is the commonly understood meaning of Mahayana Buddhism, i.e., from gradual cultivation to sudden enlightenment, and then, from sudden enlightenment to perfect enlightenment. Bencai's view that, through the realization that there is no other Buddha outside the mind, meditation practice is improved, and through cultivation, the ultimate truth is realized, is consistent with that of Zongmi. As Huineng accumulated many experiences of meditation practice in past lives, he could immediately achieve enlightenment in this life. At the end of the text, the epilogue exhorts practitioners again emphasizing, "For those who practice Chan, seated meditation is the most important thing", 學道之人，坐禪為要. Through this, he attaches importance to seated meditation.

Overall, Bencai's "Seated Meditation Etiquette" clearly states at the beginning that seated meditation is useful for regulating the mind, and the last passage reemphasizes the importance of seated meditation, guiding meditators from "stopping" to "seeing", and it reiterates the order of practicing seated meditation. There is a clear order of meditation practice from "unawareness" to awakening one's own mind through gradual cultivation is a Buddha, and there is no Buddha outside of one's own mind. From the "beginning enlightenment" to the following awakening and gradual cultivation to prove "original enlightenment" stemming from cultivation as well, a clear order is followed. This idea, which attaches importance to the order of practice, is consistent with that of Qingyuan.

## 5. Conclusions

The literature review found that the Chan masters in the Tang Dynasty taught their disciples by providing guidance with various physical actions or simple words, and the disciples were immediately enlightened. As there was no fixed method of guidance, it depended purely on the basic characteristics and circumstances of the objects, and the fundamental characteristics of the objects were adjusted according to the environment of the times. This study reviewed how Huineng and Mazu opposed physical seated meditation and attached importance to life practice during the Tang Dynasty. Subsequently, the Chan masters of the Linji school also opposed physical seated meditation. Moreover, this study performed textual criticism regarding two aspects, namely, the practice and theory of seated meditation, and found that the attitude of Chan masters in the Linji school toward seated meditation during the Song Dynasty differed from that of the Linji Patriarch in the Tang Dynasty, during which time physical seated meditation was opposed. In terms

of practice, although the Huanglong and Yangqi schools seemed to inherit the Linji school in concept and opposed seated meditation, their main way of practice was to investigate Chan Koan. Actual investigations revealed that the Linji masters in the Song Dynasty had more experience of seated meditation in their practice.

In terms of meditation theory, both Qingyuan's "Seated Meditation Inscription" and Bencai's "Seated Meditation Etiquette" emphasize that deep meditation can indeed be extended to depart from seated meditation, thus eliminating the difference between sitting and no longer sitting; therefore, there is no need to be obsessed with physical seated meditation. In the past, the Linji school's discourses mostly emphasized that, after meditation, there was no difference between sitting and no longer sitting, but they neglected to explain how to reach the threshold of samadhi. Consequently, the attitude of Qingyuan and Bencai toward seated meditation had already changed from that of Mazu's time and his disciple Erhu Dayi's "Seated Meditation Inscription", which opposed physical seated meditation. They both affirmed that seated meditation is the basis of enlightenment to self-nature, and they both valued the concept of achieving enlightenment through gradual cultivation, a concept that differed from that of Yuanwu Keqin and Dahui Zonggao, who were the prevailing authority in the Linji school at the time and valued the Koan-based Chan style. They intended to return to seated meditation.

Furthermore, the Linji masters in the Northern Song Dynasty developed the method of referring to Koan or Kanhua Chan as a method of meditation, and it already included the concept of following the order of meditation practice to achieve sudden enlightenment. This method is equivalent to returning to the practice path criticized by Huineng to achieve liberation from meditation and turning to the process of gradual accumulation before sudden enlightenment to achieve the ultimate sudden enlightenment. In other words, this method is the same as the order of meditation proposed by Qingyuan and Bencai, which meant seated meditation is essential basic training for meditation. Thus, the combination of Huatou and the concept of gradual cultivation became the mainstream of the Linji Chan practice from the Southern Song Dynasty.

**Funding:** This research was funded in part by Taiwan National Science and Technology Council.

**Institutional Review Board Statement:** Not applicable.

**Informed Consent Statement:** Not applicable.

**Data Availability Statement:** Not applicable.

**Conflicts of Interest:** The author declares no conflict of interest.

## Abbreviations

| | |
|---|---|
| CBETA | Taipei: Chinese Buddhist Electronic Text Association. Available online: http://www.cbeta.org/index.htm (accessed on 11 July 2023). |
| T | *Taishō shinshū daizōkyō* 大正新脩大藏經 Vols. 1–55, 85. Tokyo: Daizo Shuppansha, 1924–1935. Popular Edition in 1988 |
| X | *Manji Shinsan Dainihon Zokuzōkyō* 卍新纂大日本續藏經 Vols. 1–90. Tokyo: Kokusho Kankokai, 1975–1989. |

## Notes

[1] For example, (Cleary 1989, 2022). The former complete translation of the great Chan master Dahui's teachings is in English, while the latter helps clarify the characteristics of Linji Zen in the Tang and Song Dynasties.

[2] *The Jingde Record of the Transmission of the Lamp*, 景德傳燈錄, vol. 5, "Chan Master Nanyue Huairang" 南嶽懷讓 records how Huairang used the example of "grinding a brick into a mirror" 磨磚作鏡 to enlighten Mazu Daoyi 馬祖道一. The śramaṇa (沙門) Daoyi usually practiced seated meditation. Therefore, Huairang asked him, "Why do you practice seated meditation?" Daoyi answered, "I'd like to become Buddha". Therefore, Huairang took a brick and ground it on a stone. Daoyi asked, "Why are you grinding the brick?" The master said, "Grinding it into a mirror". Daoyi asked, "How could a brick be ground into a mirror?" The master said, "A brick cannot be ground into a mirror, so how could practicing seated meditation make you Buddha?" CBETA, T51, p. 240c.

[3]     *Chi Xiu Bai Zhang Qing Gui* 敕修百丈清規, vol. 6 "Seated Meditation to Enlightenment" 坐參: "In ancient times, one would go to investigate the abbot every night in order to seek enlightenment. Therefore, people gathered to sit and wait for the sound of the drum to visit the abbot, which is called seated meditation to enlightenment" (Dehui 1336, CBETA, T48, p. 1143b).

[4]     Shan-wu, *Quotations from Foyan Zen Master* 舒州佛眼和尚語錄, included in *"Quotations from Ku-tsun-su"* 古尊宿語錄, vol. 33, (Ze zang zhu, n.d. CBETA, X68, p. 226b-c.

[5]     The Buddha's Dhyāna is called Naga, also known as "constant Dhyāna". See Ciyi (2011). *Dictionary of Chinese Buddhist Terms*, p. 3025.

[6]     In the *Awakening of Faith in the Mahayana*. The original awareness of all living beings is called the "original enlightenment". After acquired practice, gradually remove the contamination of delusion from beginningless time and realize the origin of the innate mind, which is called the "beginning enlightenment". Achievement of the original awareness from the beginning awakening is called the "ultimate enlightenment".

[7]     The translation of the terms "stopping" and "seeing" is obtained from Thomas Cleary's book, (Cleary 1997).

[8]     Bodhisattvas before the eighth Bhumi have not yet attained freedom in the realm of true suchness. However, Bodhisattvas above the eighth Bhumi can benefit all living beings at will. This is called "the path of an-ābhoga". See Ciyi (2011). *Dictionary of Chinese Buddhist Terms*, p. 5077.

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
