# Peer review of "Transformation of the Linji School’s Perspective on Seated Meditation from Tang to Song Dynasties—From Negation to Returning of Seated Meditation"

_religions, doi:10.3390/rel14091129_

Round 1

Reviewer 1 Report

The article offers an interesting alternative view to a certain (now dated) image of Song-era chan, divided between Dahui's kanhua chan and Hongzhi's Mozhao chan, and does so by analyzing in detail not only documents written by masters themselves, but also other works such as biographies and lesser-known “lamp histories”. By doing so it offers a much more nuanced picture of the Linji school's attitude toward the supposed "rival" practice of sitting meditation.

I would surely recommend its publication. However, the value of this article is definitely diminished by a whole series of formal issues that need to be addressed before being published.

1) Greatly improve the citation system

One must consider the editorial destination: "Religions" is an international journal aimed at scholars of religions, not just specialists in particular religious traditions, in this case Chinese or Japanese. 

Therefore, I recommend that when citing a source in Chinese and Japanese, to do so by citing the original and/or in pinyin and romaji, followed then by a tentative translation of the title. As far as I can see, a lot of journals which are originally in Japanese are cited as they were English publication, for example “Journal of Indian and Buddhist Studies”, which I am pretty sure is Indogaku Bukkyōgaku Kenkyū.

I have also noticed that sometimes citations from primary sources are given in footnotes, sometimes in brackets, and sometimes (indeed very often) bibliographical indications of the translated primary sources are just absent.

I recommend always citing the original whenever there is a quotation from a primary source, even when it involves largely known phrases such as the one from Mazu and his brick. Also a welcomed addition would be the entire quotation of the original in footnotes. It has been done in previous articles of Religions dealing with textual studies of zen.

2)Clarity of style and choice of words.

In general, while sections 1, 2 and 3 are written in a fairly clear and fluent style, when the author analyzes the writings of Qingyuan and Bencai in detail, the discussion becomes difficult to follow. At times it seems that the author adopts the perspective of an insider speaking to other insiders, using theological terminology that is very technical.  Let us not forget that the publishing destination is an international journal aimed at scholars of religions, not Zen specialists, so a more careful argumentation, phrasing and use of technical terms is highly suggested.

For example, in lines 452-455 the author speaks of "fulfillment of enlightenment" and "to prove enlightenment," terms that are unclear to me as they are not accompanied by the original. They could also be technical terms specific to Zongmi (my knowledge of Chan is not that advanced). In any case, some more elucidation would help. In fact, I would have had the same problem when the author talks about "beginning of awakening" and "primitive awakening". If they had not been accompained by the original Chinese terms and the Dachengqixinlun citation, I would not have recognized them. In English publications it is usually used as translation "original enlightenment", instead of "primitive awakening".

Probably, because the author aims at using a very technical way of exposition, but not in his/her mother tongue, as a result some sentences are quite unclear. On this regard, I highly recommend having a professional proofreading by a mother-tongue proofreader.

Unclear passages are the following:

Line 195 "before they can become broaden sentient beings." The meaning is unclear.

Lines 228-230: it is not clear what is meant by "higher level of the realm," "personal habits," and "the two realms mentioned by Fanghui"

Line 340: unclear what is meant by "ideas sourced from the mind"

Line 356: it is not clear what is meant by "is the process of a method"

Line 404: it is not clear what is meant by "which is an intention related to the experience level".

line 235-239: better indicate the citation within the citation.

3) Doubts about the use of certain terms.

I am unconvinced and I don't understand the reason for the use of terms like "samatha", "vipassana", "mindfulness" when referring to Chan texts and practices.

Is it due to the fact that the corresponding Chinese terms (guan, shi, ding, nian) are found in the texts analyzed? Then why not use such terms? Or does the author consider "samatha", "vipassana" or "mindfulness" as a kind of technical scholarly terminology to denote, respectively, the meditative technique of stabilizing the mind and that of experiential discernment of the doctrinal principles of Buddhism? If the latter is the case, I'd like the author to address more in detail the choice of his/her use of this kind of terminology, especially in consideration of the fact that nowadays terms like vipassana are often used to refer to modern interpretation of theravada meditative practices.

Last but not least, would it be better to stick with Chinese and use chan instead of zen, shexin instead of sesshin, etc.? I come from Japanese Studies but I prefer reading terms in Chinese when having to do with Chinese topics.

4) Bibliography

The bibliography is quite short, which is okay since the paper involves a close reading of some primary sources, but for typical readers, there needs to be additional English-language sources cited since there have been published many works on the topics of the article. For example on note 4 a 2013 work is cited addressing the rhetorical and 'political' aims behind certain depiction of the southern and northern school, which is a topic John R. McRae has already addressed in his 2003 "Seeing through Zen", University California Press.

5) Other unclear passages

line 109: is it by any chance an error that the northern school is mentioned? I point this out because it seems to me that the southern school is mentioned, since also the Platform Sutra is mentioned.

line 148: what does it mean? that issues of power and rhetoric do not transpire from "lamp histories"?

line: 180 Shoushan Xingnian is shown without dates and it is not explicitly stated that he is a master of the Linjii school.

Probably, because the author aims at using a very technical way of exposition, but not in his/her mother tongue, as a result some sentences are quite unclear. On this regard, I highly recommend having a professional proofreading by a mother-tongue proofreader.

Author Response

I am very grateful to the reviewers who read my paper carefully and provided valuable comments, and I have completely revised the paper according to these comments. The following is my modification description:

Reviewer 1
The article offers an interesting alternative view to a certain (now dated) image of Song-era chan, divided between Dahui's Kanha chan and Hongzhi's Mozhao chan, and does so by analyzing in detail not only documents written by masters themselves but also other works
such as biographies and lesser-known “lamp histories”. By doing so it offers a much more nuanced picture of the Linji school's attitude toward the supposed "rival" practice of sitting meditation.
I would surely recommend its publication. However, the value of this article is definitely diminished by a whole series of formal issues that need to be addressed before being published.

1) Greatly improve the citation system
One must consider the editorial destination: "Religions" is an international journal aimed at scholars of religions, not just specialists in particular religious traditions, in this case, Chinese or Japanese.
Therefore, I recommend that when citing a source in Chinese and Japanese, do so by citing the original and/or in pinyin and romaji, followed then by a tentative translation of the title. As far as I can see, a lot of journals which are originally in Japanese are cited as they were English publications, for example, “Journal of Indian and Buddhist Studies”, which I am pretty sure is Indogaku Bukkyōgaku Kenkyū.
I have also noticed that sometimes citations from primary sources are given in footnotes, sometimes in brackets, and sometimes (indeed very often) bibliographical indications of the translated primary sources are just absent.

I recommend always citing the original whenever there is a quotation from a primary source, even when it involves largely known phrases such as the one from Mazu and his brick. Also a welcomed addition would be the entire quotation of the original in footnotes.

the entire quotation of the original in footnotes.

Respond:

The important citations of the whole paper have been added the original Chinese text as much as possible.Footnotes and bibliography have been added and revised, the quotes have indicated the source, except for two citations that are second citations(Line462 and Line473). The notes added from 15 to 25.

2)Clarity of style and choice of words.

In general, while sections 1, 2 and 3 are written in a fairly clear and fluent style, when the author analyzes the writings of Qingyuan and Bencai in detail, the discussion becomes difficult to follow. At times it seems that the author adopts the perspective of an insider speaking to other insiders, using theological terminology that is very technical.  Let us not forget that the publishing destination is an international journal aimed at scholars of religions, not Zen specialists, so a more careful argumentation, phrasing and use of technical terms is highly suggested.

For example, in lines 458-467 the author speaks of "fulfillment of enlightenment" and "to prove enlightenment," terms that are unclear to me as they are not accompanied by the original. They could also be technical terms specific to Zongmi (my knowledge of Chan is not that advanced). In any case, some more elucidation would help. In fact, I would have had the same problem when the author talks about "beginning of awakening" and "primitive awakening". If they had not been accompained by the original Chinese terms and the Dachengqixinlun citation, I would not have recognized them. In English publications it is usually used as translation "original enlightenment", instead of "primitive awakening".

Respond:

lines 464-473, revised to:

Zongmi’s “Preface to the Collection of Various Writings on the Chan Source” 禪源諸詮集都序 advocates the practice sequence of “After sudden enlightenment, there is still need for gradual cultivation.” From Zongmi’s perspective, "Enlightenment" has two levels: “If one practices because of awakening, this is enlightened understanding; and then if one becomes enlightened through practice, this is enlightenment. 若因悟而修,即是解悟; 若因修而悟,即是證悟。 All the points above are only limited to this life, if we return to the accumulation of many lives in the past, it is all enlightenment through gradual cultivation, not sudden enlightenment. 然上皆只約今生而論,若遠推宿世,則唯漸無頓。 Sudden enlightenment in this life is achieved through the accumulation of gradual cultivation in many past lives.” 今頓見者,已是多生漸熏而發現也。(CBETA, T48, p. 408a02.)

Line382-393:

the Qingyuan Quotations read: “If one wants to achieve the original purity of his mind, he needs to go through the process of seeking and visiting a master, studying day and night, and cultivating his mind until he achieves enlightenment. Afterward, he will realize that he never lost his pure mind, even when he had not aroused his mind. The Patriarch Maming 馬鳴 called this ‘beginning enlightenment’ 始覺, which is ‘original enlightenment,’ 本覺 as there is no difference between beginning enlightenment and original enlightenment, and they are collectively called the ‘ultimate enlightenment’ 究竟覺.”

Add notes23

Song Dynasty, Edited by Shan-wu, Quotations from Foyan Chan Master 舒州佛眼和尚語錄, included in "Gu Zunsu Quotations" 古尊宿語錄, vol. 33, CBETA, X vol. 68, p. 217c01. In the Awakening of Faith in the Mahayana. The original awareness of all living beings is called the “original enlightenment.” After acquired practice, gradually remove the contamination of delusion from beginningless time and realize the origin of the innate mind, which is called the “beginning enlightenment.” Achievement of the original awareness from the beginning awakening is called the “ultimate enlightenment.”

Line393-397:

Qingyuan quoted Awakening of Faith in the Mahayana 大乘起信論 to explain the process of achieving the state of the original pure mind from the concept; this means “beginning enlightenment” will eventually lead to “original enlightenment.” Apparently, Qingyuan agreed that enlightenment can only be achieved through the process of gradual accumulation. It is not achieved all at once, but through stages of meditation.

Probably, because the author aims at using a very technical way of exposition, but not in his/her mother tongue, as a result some sentences are quite unclear. On this regard, I highly recommend having a professional proofreading by a mother-tongue proofreader.

Unclear passages are the following:

Lines 206 "before they can become broaden sentient beings." The meaning is unclear.

Lines 206-210 Revised to:

Jia Tai Pu Lamp Records嘉泰普燈錄, Volume 3, records that Huanglong Huinan once said, “Those who become monks must express the ambition of a man of great courage, cut off the two ends, and return home to sit firmly, so as to open the door to benefit human beings.” In other words, he advocated that monks must first achieve seated meditation before helping sentient beings.

Lines 234: it is not clear what is meant by "higher level of the realm," "personal habits," and "the two realms mentioned by Fanghui"

Lines234-238 Revised to:

Yangqi Fanghui Monk Quotations records that he once went to the hall and said, “Stick to sitting until the universe disappears, the world will be dark; letting go of the obsession, the rain will stop, and the sun will shine again. Even so, there are still habits that have not been eliminated.”14 坐斷乾坤,天地暗黑;放過一著,雨順風晴。雖然如是,俗氣未除在。

Added notes 14:

Quotations from Monk Yangqi Fanghui 楊岐方會和尚語錄, CBETA, T vol. 47, p. 640c11.

Lines 245-248 revised to:

he naturally let go of his obsession with physical seated meditation and entered a higher level of the realm. At this moment, while his inner meditation was stable, there were still personal inertia and habits. Therefore, the two realms mentioned by Fanghui were not opposite, and his words should not be interpreted as negating seated meditation.

line 235-239: better indicate the citation within the citation.

Revised to:

Yuanwu Keqin once said, “After the Buddha was born, he merely taught to cessation knowing.” Afterwards, you will achieve the goal of “walking is walking, sitting is sitting, wearing clothes is wearing clothes, and eating is eating, which is similar to a bright mirror where everything is reflected as it is; without the awareness of discrimination, self-nature can be seen everywhere in daily life.” 15

Add notes 15:Quotations from Chan Master Yūan-Wu Fo-Kuo圓悟佛果禪師語錄 Volume 13,CBETA, T47, no.1997, p.773b07-b19.

Line 351-358: unclear what is meant by "ideas sourced from the mind"

Revised to:

“From the very beginning sat upright, the mind was full of distractions" 初心鬧亂. This refers to a delusional mind that has not been trained and is without awareness. After a long time, “the six senses will no longer respond to the external environment,” 虛閑六門, Even when the senses engage disparate thoughts, these thoughts themselves remain impermanent. As long as there is the awareness that thoughts originate from your own mind, the reactions of the six senses will gradually stop, and the thoughts will gradually become clear, i.e., the “mind becomes unobstructed,” 心心無礙, which transcends the binary system.  

Line 372: it is not clear what is meant by "is the process of a method"

Revised to:

seated meditation is a method, not an end.

Line 418-420: it is not clear what is meant by "which is an intention related to the experience level".

Revised to:

He further pointed out that the focus of seated meditation is to adjust the "mind" from contemplating the present mind in reality, which is a thought biased toward the experiential level.

3) Doubts about the use of certain terms.

I am unconvinced and I don't understand the reason for the use of terms like "samatha", "vipassana", "mindfulness" when referring to Chan texts and practices.

Is it due to the fact that the corresponding Chinese terms (guan, shidingnian) are found in the texts analyzed? Then why not use such terms? Or does the author consider "samatha", "vipassana" or "mindfulness" as a kind of technical scholarly terminology to denote, respectively, the meditative technique of stabilizing the mind and that of experiential discernment of the doctrinal principles of Buddhism? If the latter is the case, I'd like the author to address more in detail the choice of his/her use of this kind of terminology, especially in consideration of the fact that nowadays terms like vipassana are often used to refer to modern interpretation of theravada meditative practices.

Respond:

Line90-96:

The Fourth Patriarch Daoxin 四祖道信 attached great importance to the accumulation of meditation skills. The Jingde Record of the Transmission of the Lamp景德傳燈錄 mentions that he “practiced Sesshin and never lay down to sleep for 60 years.” He advocated the following in The Suitable Method of Attaining Contemplation and Enlightenment of Mind入道安心要方便法門: “If one only practices Samatha for a long time, his mind will easily feel dull; if one only practices Vipassana for a long time, his mind will easily become distracted.” 常住於止,心則沉沒;久住於觀,心則散亂。

Line99-101:

Daoxin emphasized that as there is no other Buddha outside the mind, it is most important to cultivate the “mind.”

Line107-108:

the cultivation of mind in daily life has become the focus of Chan practitioners’ practice

Line412:The above content pays particular attention to the practice of “Samatha”.

Revised to:The above step is similar to practicing "stopping."

Line416:which pays particular attention to the outcome of practicing “Vipassana”.

Revised to:The above part is similar to practicing "seeing."

Line405-425:

Revised to:

The beginning of the text explains the level transformation of seated meditation and mind cultivation: When performing seated meditation, one should first “correct his thoughts” 端心正意 and adjust his motivation; the body posture should be “stacking feet and sitting cross-legged” 疊足加趺. Then, he should “watch and listen to his inner world,” 收視反聽, that is, withdraw his gaze from the outside to the inside to maintain awareness. In addition, he should “stay away from drowsiness and restlessness,” 沉掉永離 that is, any distracting thoughts should be ignored, and a state of mental tranquility should be maintained. The above step is similar to practicing "stopping." Second, after mental concentration and stability is achieved, use the calm mind to maintain awareness, in which state one is “aware of the present mind and reflecting on the present mind” 知坐是心,及返照是心. “This mind is empty, aware, still, and clear,” 此心虗而知、寂而照 and does “not fall into the state of extinguishing and eternity.” 不墮斷常 The above part is similar to practicing "seeing." Therefore, Bencai’s guidance on seated meditation includes the concept of first practicing stopping and then practicing seeing.24 He further pointed out that the focus of seated meditation is to adjust the "mind" from contemplating the present mind in reality, which is a thought biased toward the experiential level. Being aware of and reflecting on the present mind are similar to what is described in Qingyuan’s “Seated Meditation Inscription”: “Use one’s own mind to reflect on it again” 還用自心,反觀一遍. This “mind” is peaceful and always able to achieve enlightenment, and when enlightenment is achieved, one can always maintain awareness, i.e., can see the original Buddha nature.

Add note24:The translation of the two terms "Stopping" and "seeing" is quoted from Thomas Cleary 's book, Stopping and Seeing:A Comprehensive Course in Buddhist Meditation, shambhala publications press, 1997.

Last but not least, would it be better to stick with Chinese and use chan instead of zen, shexin instead of sesshin, etc.? I come from Japanese Studies but I prefer reading terms in Chinese when having to do with Chinese topics.

Respond:

The full text is changed from "zen" to "chan", and important citations plus original Chinese text.

4) Bibliography

The bibliography is quite short, which is okay since the paper involves a close reading of some primary sources, but for typical readers, there needs to be additional English-language sources cited since there have been published many works on the topics of the article. For example on note 6 a 2013 work is cited addressing the rhetorical and 'political' aims behind certain depiction of the southern and northern school, which is a topic John R. McRae has already addressed in his 2003 "Seeing through Zen", University California Press.

This question is related to another question--

line 159: what does it mean? that issues of power and rhetoric do not transpire from "lamp histories"?

Respond:

lina159-164 revised to:

American scholar John R. McRae pointed out that the discourses on Chan history in both the northern and southern schools conceal strong school narratives and rhetorical power, which are irrelevant to literature. To a certain extent, the southern school’s opposition to seated meditation may have been a development strategy to differentiate itself from the northern school.9

note 9, revised to:

John R. Mcrae, ”Looking at Lineage: A Fresh Perspective on Chan Buddhism”, Seeing through ZenEncounter, Transformation, and Genealogy in Chinese Chan Buddhism, University California Press, 2004, p.1-21.

5) Other unclear passages

line 122: is it by any chance an error that the northern school is mentioned? I point this out because it seems to me that the southern school is mentioned, since also the Platform Sutra is mentioned.

Respond:

Thanks a lot! line 122-128, It indeed is southern school.

The southern school, which transcended the obsession with the seated meditation of the body, eliminated the formal boundary between sitting up and sitting down, and emphasized awareness of the mind, criticized the northern school’s seated meditation, which attaches importance to the outer appearance, stating, “Some masters teach seated meditation by asking Buddhists practitioners not to move their body, so that their mind can be purified. This is the inversion of obsession.”4

Add notes4:

Dunhuang version, recorded by Fahai, Platform Sutra, CBETA, T vol. 48, p. 339, a04–09.

line: 195 Shoushan Xingnian is shown without dates and it is not explicitly stated that he is a master of the Linjii school.
Respond:
Third chapter: The seated meditation practice of Chan masters in the Linji school in the Song Dynasty, It's all about the Chan masters of the Linji sect in the Song Dynasty.
Line195, To add note: Shoushan Xingnian 首山省念(926-993).

Comments on the Quality of English Language
I highly recommend having professional proofreading by a mother-tongue proofreader.
Respond:
The whole thesis has been proofread by a mother-tongue proofreader again.

Reviewer 2 Report

In this paper, the author explores the development of seated meditation within the Linji Zen school, specifically examining the Tang and Song Dynasties. While the topic is intriguing, there are areas that require further elaboration to strengthen the claims presented.

Firstly, the reliance on Zen masters' sayings as the primary evidence for the argument is valuable. However, to enhance the paper's credibility, additional sources from the historical context should be incorporated. Considering the context and background in which these conversations occurred is essential in substantiating the author's claims effectively.

Moreover, the distinction between the Tang and Song Dynasties' approaches to seated meditation is well-highlighted. However, the paper could benefit from a clearer discussion on how the method of seated meditation evolved during the Song Dynasty within the Linji school. Presenting a more explicit and logical flow of the changes in the practice during this period would make the paper more convincing.

Additionally, to provide a comprehensive analysis, the author should include recent relevant research in this field. Referring to works like Albert Welter's "The Linji lu and the Creation of Chan Orthodoxy" (2008) and Thomas Cleary's research would add depth and diverse perspectives to the paper.

In conclusion, this paper offers valuable insights into the topic of seated meditation in the Linji Zen school. However, to enhance its scholarly merit, further supplementation of historical evidence, a clearer discussion on the evolution during the Song Dynasty, and incorporation of more recent studies are recommended. Addressing these aspects will strengthen the overall quality and impact of the research.

Author Response

I am very grateful to the reviewers who read my paper carefully and provided valuable comments, which I have completely revised the paper according to these comments. The following is my modification description:

Reviewer 2

In this paper, the author explores the development of seated meditation within the Linji Zen school, specifically examining the Tang and Song Dynasties. While the topic is intriguing, there are areas that require further elaboration to strengthen the claims presented.

Firstly, the reliance on Zen masters' sayings as the primary evidence for the argument is valuable. However, to enhance the paper's credibility, additional sources from the historical context should be incorporated. Considering the context and background in which these conversations occurred is essential in substantiating the author's claims effectively.

Moreover, the distinction between the Tang and Song Dynasties' approaches to seated meditation is well-highlighted. However, the paper could benefit from a clearer discussion on how the method of seated meditation evolved during the Song Dynasty within the Linji school. Presenting a more explicit and logical flow of the changes in the practice during this period would make the paper more convincing.

Additionally, to provide a comprehensive analysis, the author should include recent relevant research in this field. Referring to works like Albert Welter's "The Linji lu and the Creation of Chan Orthodoxy" (2008) and Thomas Cleary's research would add depth and diverse perspectives to the paper.

In conclusion, this paper offers valuable insights into the topic of seated meditation in the Linji Zen school. However, to enhance its scholarly merit, further supplementation of historical evidence, a clearer discussion on the evolution during the Song Dynasty, and incorporation of more recent studies are recommended. Addressing these aspects will strengthen the overall quality and impact of the research.

Respond:

  1. How the method of seated meditation evolved during the Song Dynasty within the Linji school.

In chapter two:Development of Linji School’s seated meditation attitude before the Tang Dynasty, I discuss from Chan School’s first monk, Bodhidharma, to northern school and Southern schools had different attitudes toward seated meditation, until Nanyue Huairang, Linji Yixuan’s attitude toward seated meditation. I hope through this discussion, readers can understand the development of Chan meditation before the Tang Dynasty.

  In chapter three:The seated meditation practice of Chan masters in the Linji school in the Song Dynasty, It mainly explains the historical development of Linji Zong in the Song Dynasty from Fenyang Shanzhao (947-1024) to Yang Qifanghui (992-1049) and Huanglong Huinan (1002-1069) Chan meditation. In order to highlight that the fourth chapter is also Linji Zong's Foyan Qingyuan and f Foxin Bencai are different from the past Linji's Chan meditation tradition.

  1. Incorporation of more recent studies are recommended.

Respond:

Thanks a lot!

In first chapter “Introduction”, Add the following content:

Line35add:

Regarding the classics of Chan Buddhism, Chan masters’ instructions regarding Chan practice and Chan master-disciple wisdom dialogues presented in the series of translations by the outstanding contemporary Buddhist translator and scholar Thomas Cleary are worthy of reference.1

Note1:For example, Thomas Cleary, Treasury of the Eye of True Teaching: Classic Stories, Discourses, and Poems of the Chan Tradition, Shambhala Publications Press, 2022; Zen Lessons: The Art of Leadership, Shambhala Publications Press, 1989. The former complete translation of the great Chan master Dahui's teachings is in English, while the latter helps clarify the characteristics of Linji Zen in the Tang and Song Dynasties.

Line52 add:

Albert Welter, a contemporary Chan researcher, has pointed out that our impression of the Linji school in the Tang Dynasty reflects the image of a Chan master constructed through the "Linjilu," which was edited in the Song Dynasty.2 We can only consider the historical background of the text as much as possible to understand the development of and changes in the Linji school’s concept of meditation in the Tang and Song Dynasties, as presented in the text.

Note2:Albert Welter, The Linji lu and the Creation of Chan Orthodoxy: The Development of Chan's Records of Sayings Literature, Oxford university press, 2008.

Round 2

Reviewer 2 Report

The author has incorporated the corrections I previously requested, and I have acknowledged that the paper's quality has improved as a result.

Author Response

Thanks a lot.